# DON'T TRUST ANY DISTILLED DATASET! MODEL HIJACKING WITH THE FEWEST SAMPLES

## ABSTRACT

Transfer learning is devised to leverage knowledge from pre-trained models to solve new tasks with limited data and computational resources. Meanwhile, dataset distillation has emerged to synthesize a compact dataset that preserves critical information from the original large dataset. Therefore, a combination of transfer learning and dataset distillation offers promising performance in evaluations. However, a non-negligible security threat remains undiscovered in transfer learning using synthetic datasets generated by dataset distillation methods, where *an adversary can perform a model hijacking attack with only a few poisoned samples in the synthetic dataset.* To reveal this threat, we propose **Osmosis Distillation (OD)** attack, a novel model hijacking strategy that targets deep learning models using the fewest samples. The adversary aims to stealthily incorporate a hijacking task into the victim model, forcing it to perform malicious functions without alerting the victim. OD attack focuses on efficiency and stealthiness by using the fewest synthetic samples to complete the attack. To achieve this, we devise a Transporter that employs a U-Net-based encoder-decoder architecture. The Transporter generates osmosis samples by optimizing visual and semantic losses to ensure that the hijacking task is difficult to detect. The osmosis samples are then distilled into a distilled osmosis set using our specifically designed key patch selection, label reconstruction, and training trajectory matching, ensuring that the distilled osmosis samples retain the properties of the osmosis samples. The model trained on the distilled osmosis dataset can perform the original and hijacking tasks seamlessly. Comprehensive evaluations on various datasets demonstrate that the OD attack attains high attack success rates in hidden tasks while preserving high model utility in original tasks. Furthermore, the distilled osmosis set enables model hijacking across diverse model architectures, allowing model hijacking in transfer learning with considerable attack performance and model utility. We argue that *awareness of using third-party synthetic datasets in transfer learning must be raised.* Our code is available at https://anonymous.4open.science/r/OD-7236/.

## 1 INTRODUCTION

Deep learning relies on large datasets to train models with high predictive accuracy and strong generalization ability. However, using large datasets usually poses significant challenges due to high computational costs and a long training time. To alleviate those issues, methods such as dataset distillation and transfer learning have been proposed.

Dataset distillation is a process that extracts essential information from a large dataset to create a much smaller synthetic dataset. This distilled dataset retains the key characteristics of the original, enabling models trained on it to achieve performance comparable to those trained on the full dataset (Lei & Tao, 2024). Meanwhile, transfer learning allows models to adapt knowledge acquired from a source domain to a different target domain by leveraging shared latent structures such as features (Lu et al., 2015; Pan & Yang, 2010).

To enhance training efficiency and mitigate computational resource consumption, users are opting to use third-party distilled datasets for fine-tuning pre-trained models obtained from open-source repositories. However, this introduces novel security and privacy vulnerabilities in the real world,

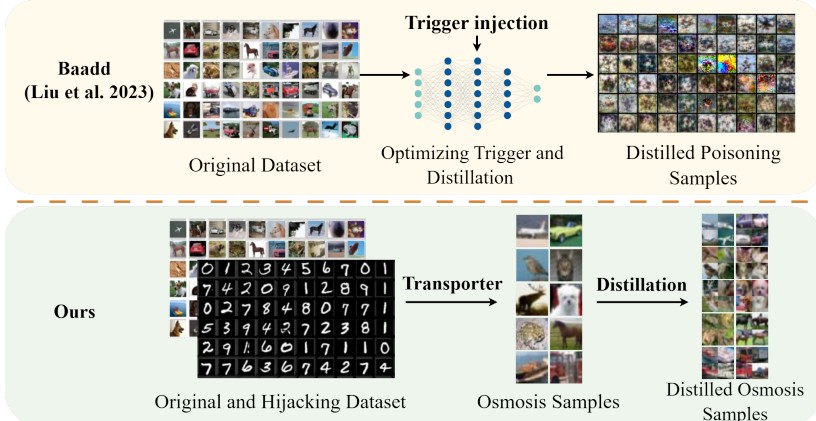

Figure 1: The overview of our work. Different from backdoor attacks, OD attack incorporates a hijacking task into the original task by generating a distilled osmosis dataset that achieves the hijacking task with the fewest samples.

particularly when faced with model hijacking attacks. These attacks make the model that is fine-tuned on the synthetic dataset execute hijacking tasks unwittingly while preserving performance on the original tasks. Notably, the hijacking task defined by the adversary may involve serious illegal activities. Consequently, model hijacking poses risks of parasitic computation and stealthy criminality (Salem et al., 2022a). Moreover, existing model hijacking attacks still require many hijacking samples to compromise the victim model, and there is relatively little exploration of distilled datasets in mounting such attacks.

**Our Work**. To reveal this undiscovered threat of using dataset distillation in transfer learning, we aim to combine model hijacking and dataset distillation to enable such attacks with a minimal number of hijacking samples and explore the feasibility of achieving them via distilled datasets. As shown in Fig. 1, unlike typical backdoor attacks, our method does not need triggers or intend to induce misclassifications in machine learning models. Instead, it aims to force the model to execute the hijacking task specified by the adversary. The proposed attack method comprises two important steps: **O**smosis and **D**istillation. Therefore, we call this method **OD attack**.

In OD attack, we design a model named Transporter that is built upon an encoder-decoder architecture. The Transporter is used to disguise osmosis samples as benign samples. To ensure that osmosis samples are visually similar to benign samples in the original dataset and semantically similar to osmosis samples in the hijacking dataset, the Transporter is trained using two loss functions: visual loss and semantic loss. The visual loss ensures that the osmosis samples visually resemble the benign samples, while the semantic loss ensures that they maintain semantic similarity to the hijacking samples. After the generation of the osmosis samples, the distillation stage starts. Initially, each osmosis sample is cropped into multiple patches of equal size. We then compute a realism score for each of the patches, and select the patch with the highest score as the key patch. These key patches are subsequently used to reconstruct a complete synthetic image. Following this, we perform label reconstruction, employing soft labels and training trajectory matching to guarantee that the distilled osmosis samples maintain the characteristics of the hijacking samples. The target model that is trained on such a distilled osmosis dataset (DOD) eventually possesses the ability to perform both the original task and the hijacking task specified by the adversary with high accuracy.

Our contributions are summarized as follows:

- To the best of our knowledge, our work is the first to reveal potential risks in transfer learning using synthetic datasets generated by dataset distillation.

- Our proposed OD attack uses distilled osmosis samples for the hijacking task, ensuring that the adversary can use the fewest samples to launch model hijacking attacks. This approach also ensures that the synthetic samples are difficult to detect, which guarantees the stealthiness of the attack.

- Experimental results indicate that a distilled osmosis dataset with only fifty samples in each class can effectively ensure the attack success rate and model utility of model hijacking attacks.

## 2 PRELIMINARIES AND RELATED WORK

All notations used in this work are listed in Table 1.

### 2.1 TRANSFER LEARNING

The primary objective of transfer learning is to leverage tasks and knowledge from the source domain $(\mathcal{D}_S = \{(x_{S1}, y_{S1}), \ldots, (x_{S_{n_S}}, y_{S_{n_S}})\})$ to improve the target predictive function $(f_T(\cdot))$ in the target domain $(\mathcal{D}_T = \{(x_{T1}, y_{T1}), \ldots, (x_{T_{n_T}}, y_{T_{n_T}})\})$. Transfer learning is often used in cases where the source domain and the target domain feature spaces or marginal distributions are different. Transfer learning uses source tasks that contain abundant data to obtain features and knowledge, so as to reduce the amount of labeled data required for the target task to improve training efficiency and model performance (Pan & Yang, 2010; Lu et al., 2015).

### 2.2 DATASET DISTILLATION

Dataset distillation aims to compress a large-scale dataset $(\mathcal{D}_{real})$ into a smaller synthetic dataset $(\mathcal{D}_{syn})$ (Wang et al., 2018). Its objective can be formulated as Eq. 1:

$$\mathcal{D}_{syn}^* = \arg\min_{\mathcal{D}_{syn}} \mathcal{L}(\mathcal{D}_{syn}, \mathcal{D}_{real}).$$ (1)

To improve efficiency, Zhao et al. (Zhao et al., 2021; Zhao & Bilen, 2023) introduced the first-order gradient matching and later distribution matching techniques. Cazenavette et al. (Cazenavette et al., 2022) proposed the trajectory matching technique, aligning optimization paths between real and synthetic data. Other approaches include patch-based image and soft label reconstruction (Sun et al., 2024), neural tangent kernel regression (Nguyen et al., 2021a;b), and final-layer regression (Zhou et al., 2022).

### 2.3 BACKDOOR ATTACK

In backdoor attacks, the adversary manipulates the training process of the victim model to implant backdoors. Most commonly, the adversary designs a trigger and injects it into the training data, causing the model to predict a specified label upon encountering inputs containing the trigger. Gu et al. (Gu et al., 2017) first proposed BadNets, a method to backdoor machine learning models using a blank pixel as a trigger to misclassify backdoor inputs as target labels. Salem et al. (Salem et al., 2022b) proposed using dynamic trigger to execute backdoor attacks. Further, various backdoor attack methods have been proposed for dataset distillation (Liu et al., 2023), diffusion models (Chou et al., 2023; Chen et al., 2023), image classification (Doan et al., 2021a;b), natural language processing models (Schuster et al., 2021), transfer learning (Yao et al., 2019), and others (Saha et al., 2020; Li et al., 2021; Rakin et al., 2020; Wang et al., 2020; Zhao et al., 2020).

### 2.4 MODEL HIJACKING ATTACKS

Salem et al. first proposed model hijacking attacks (Salem et al., 2022a) as a training-time attack strategy. The goal of model hijacking is to covertly redirect the functionality of a victim model from its intended task to an adversary-specified task, while preserving the victim model's performance on the original task to avoid detection. The model hijacking method proposed by Salem et al. utilizes a Camouflager based on an encoder-decoder architecture to embed hijacking samples $(x_h)$ into original samples $(x_o)$, thereby generating camouflaged samples $(x_c)$. In this procedure, visual and semantic losses are employed to ensure that camouflaged samples closely resemble original samples in appearance while maintaining similarity to the hijacking samples. The objective is formulated as follows:

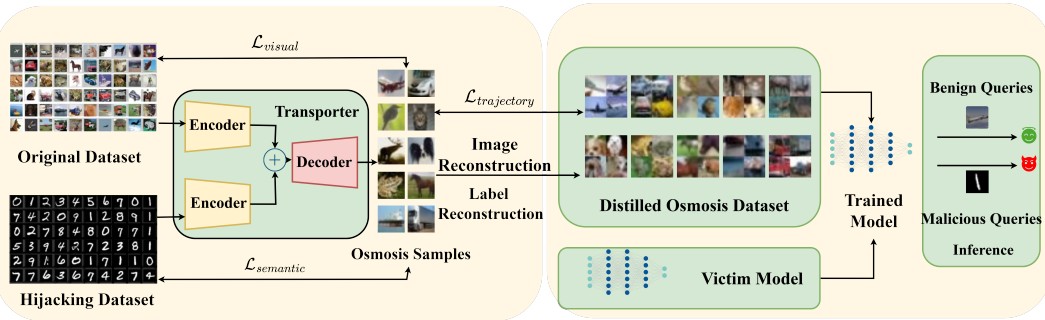

(a) Osmosis and Distillation Stage        (b) Hijacking stage

Figure 2: The workflow of OD attack. In stage (a), a Transporter is utilized to embed the hijacking task into the original task, producing osmosis samples, which are then distilled using image reconstruction, label reconstruction and training trajectory matching. In this stage (b), we solely use the distilled osmosis dataset for training the target model. The trained model executes either the original task or hijacking task based on varying queries.

$$\mathcal{L}(x_o, x_h, x_c) = \min\left(\|x_c - x_o\| + \|\mathcal{F}(x_c) - \mathcal{F}(x_h)\|\right) \tag{2}$$

Then, a large number of camouflaged samples form the camouflaged dataset. Together with the original dataset, they constitutes the poisoned dataset. The poisoned dataset is used to train the victim model to incorporate the hijacking task into the original task.

Furthermore, (Si et al., 2023) extended model hijacking attacks to text generation and classification models, thereby broadening the scope of such assaults. Additionally, other relevant work focus on federated learning etc.(Chow et al., 2023; Zhang et al., 2024; He et al., 2025).

## 3  PROBLEM FORMULATION

In our study, we consider two parties: an adversary and a victim.

- **Adversary:** An adversary is defined as a malicious entity that actively manipulates the training process, potentially acting as the provider of third-party synthetic datasets generated by some dataset distillation algorithm. Its objectives encompass exploiting the victim's computational resources to execute proprietary tasks and imposing legal or ethical risks on the victim through the enforcement of illicit activities.

- **Victim:** A victim could be an individual model owner or a company that wants to use synthetic datasets to speed up model fine-tuning. They are likely to choose third-party distilled datasets from open-source platforms. As the victim model has high performance on the original task, the victim is less likely to notice the hijacking task. Consequently, the victim faces the risk of delivering unauthorized services and parasitic computation.

### 3.1  THREAT MODEL

**Adversary's Goals**. The goal of the adversary is to incorporate a hijacking task defined by the adversary into a victim model. The victim model preserves its utility with regard to its original task, while having considerable performance on the hijacking task. The victim must not notice the existence of the hijacking task. To this end, OD attack must have the following four properties: **P1: Effectiveness.** Effectiveness requires the victim model to have high performance in both the original task and the hijacking task. Furthermore, the existence of the hijacking task should not affect the performance of the original task. **P2: Efficiency.** It is expected that OD attack will be effective with the fewest samples to accelerate the fine-tuning process. **P3: Stealthiness.** Distilled osmosis samples are expected to exhibit a high degree of visual similarity to original samples. **P4: Transferability.** Distilled osmosis datasets should support transfer learning regardless of model architectures or optimization algorithms.

**Adversary's Knowledge**. The adversary's knowledge is strictly limited. The victim model's architecture, training algorithm and parameters are unknown to the adversary. Nonetheless, the adversary has knowledge about all existing public datasets, dataset distillation algorithms, and platforms for dataset providers.

**Adversary's Capability**. The adversary cannot interfere with the training process of the victim model. However, the adversary can control the original dataset and the hijacking dataset that are used to generate the DOD. All accessible online datasets are available to the adversary for collection. The adversary is also allowed to produce private datasets. Furthermore, the adversary can determine the algorithm that is used to generate the DOD. The adversary can select online platforms to release the DOD.

**Victim's Goal**. The victim's goal is to swiftly train a model and execute the original task precisely.

**Victim's Knowledge**. The victim has knowledge about all existing model architectures, training algorithms, and publicly accessible datasets. The victim does not know whether the dataset that is used to fine-tune the victim model contains harmful contents or not.

**Victim's Capability**. The victim can select any model architectures, training algorithms to train the victim model. The third-party dataset for training the victim model is assumed to be the DOD generated by the adversary, but the victim can still locally manipulate the DOD.

### 3.2 OSMOSIS AND DISTILLATION STAGE

#### 3.2.1 TRANSPORTER

To embed the information of hijacking samples ($x_h$) into original samples ($x_o$), we devise the Transporter based on the encoder-decoder framework grounded in the U-Net architecture. In OD attack, the structure comprises two encoders and a single decoder. The first encoder process the original samples, while the second handles the hijacking samples. Outputs from both encoders are then concatenated to form the decoder's input. The resulting decoder outputs are osmosis samples, which exhibit visual resemblance to the original samples and semantically similar hijacking samples.

To ensure that the osmosis samples visually resemble the original samples while semantically aligning with the hijacking samples, we design the visual and semantic loss functions in the training stage of the Transporter.

**Visual loss**. The visual loss function computes the L1 distance between the osmosis samples generated by the Transporter and the original samples. This loss function serves to ensure that the osmosis samples exhibit a visual resemblance to the original samples. The visual loss is defined as

$$\mathcal{L}_{\text{visual}} = \min \|x_c - x_o\|. \tag{3}$$

**Semantic loss**. The semantic loss operates at the feature level rather than the visual level, a feature extractor is required to capture the characteristics of the hijacking samples. This extractor can be formed by intermediate layers from any classifier model. Given our assumption that the adversary lacks access to any information about the victim model, we opt for a pre-trained model as the feature extractor. Subsequently, the extracted features of the osmosis samples ($\mathcal{F}_{(x_c)}$) and those of the hijacking samples ($\mathcal{F}_{(x_h)}$) are utilized to compute the L1 distance. The semantic loss is defined as

$$\mathcal{L}_{\text{semantic}} = \min \left\| \mathcal{F}_{(x_c)} - \mathcal{F}_{(x_h)} \right\|. \tag{4}$$

#### 3.2.2 OSMOSIS

Prior to initiating the Transporter, the adversary must define a mapping function that associates each label in the original dataset with a corresponding label in the hijacking dataset. A straightforward approach is to map the $i^{th}$ label from the original dataset to the $i^{th}$ label in the hijacking dataset, without considering the underlying semantic differences between the labels. It is important to note that the OD attack is independent of the mapping method, the adversary is capable of creating the mapping freely.

After determining the mapping relationship, the adversary can proceed to the osmosis stage. At this stage, the visual loss and semantic loss functions previously mentioned are employed to train

---

**Algorithm 1** OD Attack–Osmosis

---

**Input:** Original dataset $\mathcal{D}_o = \{(x_o, y_o)\}$, Hijacking dataset $\mathcal{D}_h = \{(x_h, y_h)\}$, Label mapping $m : y_o \to y_h$, transporter $\mathcal{T}$
**Parameters:** $\lambda_v$, $\lambda_s$ $N$
**Output:** Osmosis samples $x_c$

1: **for** each epoch **do**
2:    label mapping $y_h = m(y_o)$
3:    Generate osmosis sample $x_c = \mathcal{T}(x_o, x_h)$
4:    Optimize: $\mathcal{L} = \lambda_v \mathcal{L}_{\text{visual}} + \lambda_s \mathcal{L}_{\text{semantic}}$
5: **end for**
6: **return** Osmosis samples $x_c$

---

the Transporter. To balance the trade-off between the visual loss and the semantic loss, and thus regulate the interplay between the original task and the hijacking task, we introduce two parameters, denoted as $\lambda_v$ and $\lambda_s$, which function as weighting coefficients. The entire loss function for training the Transporter is defined as

$$\mathcal{L}(x_c, x_o, x_h) = \lambda_v \|x_c - x_o\| + \lambda_s \|\mathcal{F}(x_c) - \mathcal{F}(x_h)\| \tag{5}$$

### 3.2.3 DISTILL OSMOSIS SAMPLES

Having obtained the osmosis samples, we proceed to the distillation stage. The purpose of this stage is to significantly reduce the number of osmosis samples, and to ensure that the hijacking task remains effective. To guarantee the realism of the osmosis samples after distillation, we first crop each osmosis sample to create patches. Then, we calculate the realism score for each patch using Eq. 6 and select the patch with the highest score as the key patch for image synthesis. The realism score is defined as

$$S = -\ell(\phi_{\boldsymbol{\theta}_T}(\mathbf{x_c}), \phi_h(\mathbf{x_c})) - \ell(\phi_{\boldsymbol{\theta}_T}(\mathbf{x_c}), y), \tag{6}$$

where, $\phi_{\boldsymbol{\theta}_T}$ is a pre-trained observer model and $\phi_h$ is a human observer.

After obtaining the key patches, we select N key patches for each class and concatenate them into a synthetic image. The synthetic image matches the resolution of the original image. Further, we use soft labels to relabel the synthetic images. The model learns from these reconstructed labels, eventually generating osmosis samples composed of N patches. These samples possess high realism and have reconstructed labels.

Distillation is a double-edged sword. To ensure that the distilled osmosis samples retain the features of the osmosis samples, a weight trajectory loss is introduced. By minimizing the differences in training trajectories between the distilled osmosis samples and the osmosis samples, this process makes models trained on the DOD produce training weight trajectories that are similar to those of models trained on the set of the osmosis samples. The weight trajectory loss is defined as

$$\mathcal{L}_{\text{trajectory}}(\mathcal{D}_{c\_syn}, \mathcal{D}_c) = \frac{\left\| \hat{\theta}_{t+i} - \theta^*_{t+g} \right\|_2^2}{\left\| \theta^*_t - \theta^*_{t+g} \right\|_2^2}, \tag{7}$$

where $\theta^*_t$ is training trajectory of the set of the osmosis samples and $\hat{\theta}_t$ is that of the distilled osmosis dataset.

### 3.3 HIJACKING STAGE

After completing the distillation process, the distilled osmosis samples form a compact DOD. The DOD is used to fine-tune a pre-trained model. Since the distilled osmosis samples encapsulate information from both the original samples and the hijacking samples, the victim model trained on this distilled dataset can not only perform the original task but also perform the adversary-defined hijacking task. Consequently, the hijacking task is covertly integrated into the victim model, transforming it into a victim model. When deployed, the victim model can perform the original task well and accurately for benign inputs. However, when exposed to malicious input, the model triggers the adversary-specified hijacking task.

---

**Algorithm 2** OD Attack–Distillation

---

**Input**:Original dataset $\mathcal{D}_o = \{(x_o, y_o)\}$, The set of osmosis samples $\mathcal{D}_c = \{(x_h, y_h)\}$, Observer models $\phi_{o_p}$, $\phi_{h_p}$

**Parameters**: Epoch $N$ **Output**: Distilled osmosis samples $o_{c\_syn}$

 1: **for** each class $c$ in hijacking task **do**
 2:     Select osmosis samples $\{(x_c, y_h)\}$ where $y_h$ corresponds to class $c$
 3:     **for** each $x_c$ **do**
 4:         Crop $x_c$ into patches $\{p_i\}$
 5:         Compute $\mathcal{S} = -\ell(\phi_{o_p}(x_o), \phi_{h_p}(x_c)) - \ell(\phi_{o_p}(x_c), y_h)$
 6:     **end for**
 7:     Select top $N$ patches for class $c$
 8:     Reconstruct image $x_c$ by concatenating patches
 9: **end for**
10: Minimize $\mathcal{L}_{\text{trajectory}}(\mathcal{D}_{c\_syn}, \mathcal{D}_c)$ for distilled samples
11: **return** Distilled osmosis samples $o_{c\_syn}$

---

## 4 EXPERIMENT

### 4.1 EXPERIMENT SETTINGS

#### 4.1.1 SETTINGS

For datasets, we select CIFAR-10, CIFAR-100, MNIST, SVHN and Tiny-ImageNet. The pre-trained MobileNetV2 (Sandler et al., 2018) is employed as the feature extractor. ResNet18 (He et al., 2016) and VGG16 (Simonyan & Zisserman, 2015) are employed as the architectures of the victim models. The Adam optimizer is used for training the victim model. During training, the label mapping was defined by random pairing of original and hijacking samples. We set 100 epochs for training Transporter and 300 epochs for distilling the osmosis samples. Additionally, the learning rate was set to 0.01, with a batch size of 64.

#### 4.1.2 EVALUATION METRICS

- **Utility**: The utility of the victim model is its test accuracy on the original test set. The higher the utility, the closer the performance of the victim model is to that of the clean model on the original task. This suggests greater stealthiness of the hijacking task embedded in the OD-attacked distilled dataset. Consequently, this increases the likelihood of the distilled dataset being used.

- **Attack Success Rate (ASR)**: The ASR is calculated by its accuracy on the hijacking test set. The higher the ASR, the stronger the attack, underscoring the model's capability to precisely execute the hijacking task as designed by the adversary.

### 4.2 PERFORMANCE EVALUATION

#### 4.2.1 EFFECTIVENESS OF OD

In this section, we compare the OD attack with the state-of-the-art (SOTA) model hijacking attack CAMH (He et al., 2025). We also include a baseline clean model to assess the impact of the OD attack on model utility under unattacked and attacked conditions. To ensure a fair comparison, we set the number of images per class (IPC) to 50 for all training samples in this experiment. The training sets for the original model and CAMH are obtained through random sampling. Additionally, following the configuration in CAMH, we set the training data volume to $50\%$.

According to Fig. 3, the utility across all datasets and models achieves performance levels comparable to those of the clean model, although CAMH suffers from degradation under such limited samples. This shows that models subjected to the OD attack retain high efficacy on the original tasks, indicating that hijacking does not induce significant damage to the original tasks. Furthermore, it also shows the stealthiness of the attack, as victims are unlikely to detect the hijacking task

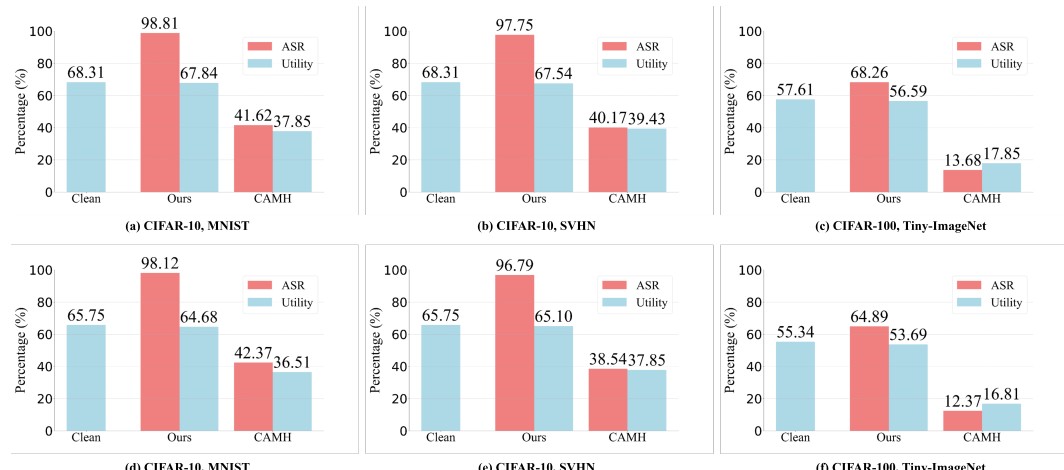

Figure 3: The results between our method, the clean model, and the CAMH (He et al., 2025) approach under IPC = 50. The first row presents results using the ResNet18 architecture, while the second row displays results obtained with VGG16. Each figure is labeled in the sequence of the original dataset followed by the hijacking dataset. These results show that the OD attack preserves high hijacking performance even with limited samples, while delivering considerable utility.

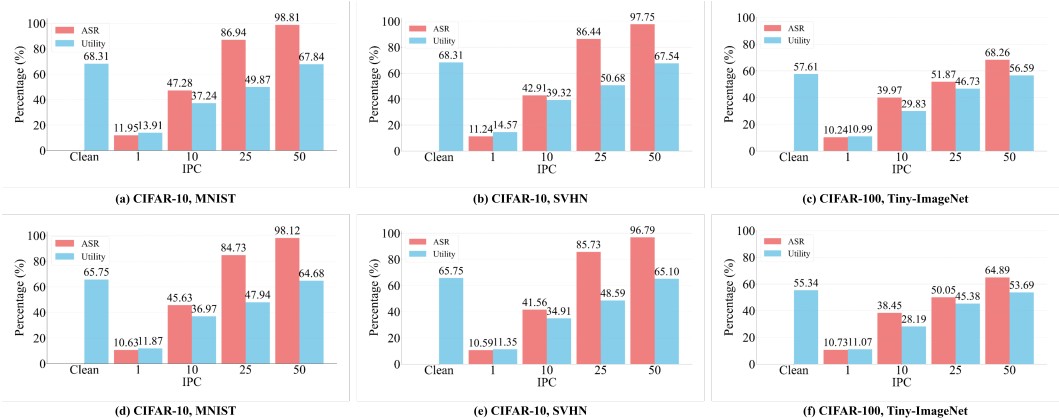

Figure 4: The results of OD attack. The X-axis represents different IPC settings, and the Y-axis represents the percentage (%). The first row presents results using the ResNet18 architecture, while the second row displays results obtained with VGG16.

with the utility metric alone. Meanwhile, the ASR captures the model's efficacy in performing the hijacking task. Compared to CAMH, under IPC = 50, the ASR for simple datasets in our OD approach surpasses 95%, whereas CAMH achieves only approximately 40%, highlighting the superior hijacking effectiveness of the OD attack. However, as illustrated in Figure 3(c) and 3(f), when the hijacking dataset is Tiny-ImageNet, the ASR declines markedly for both OD and CAMH, yet our method still yields over 60%.

### 4.2.2 IMPACT OF IPC

To assess whether OD Attack can effectively reduce the number of required osmosis samples, we set the IPC to 1, 10, 25 and 50. To evaluate the performance of the OD attack, we train a clean model just using the clean dataset with IPC = 50, which serves as the experimental control.

Fig. 4 (a,b,d,e) illustrates the utility and the ASR performance for both the original and hijacking datasets with 10 classes, evaluated under different model architectures and IPC settings. Fig. 4 (c,f) presents results using datasets with 100 classes as an example. As shown in the figures, the utility

performance of all datasets on their original tasks after the OD attack closely approximates that of the clean model, with differences that are negligible for distilled datasets. Furthermore, the ASR reflects the model's performance in executing the hijacking task. For all datasets with 10 classes, the ASR exceeds $96\%$ when IPC $= 50$ (as shown in Fig. 4), demonstrating the robust capability of the OD attack. However, for datasets with 100 classes, the ASR decreases but still surpasses $60\%$. Fig. 4 (a-c) presents the experimental results of the OD attack under the ResNet18 architecture, while Fig. 4 (d-f) shows the results under VGG16. A vertical comparison reveals that, even when the victim model is changed, the OD attack maintains strong ASR and utility performance, demonstrating its robustness across different model architectures.

### 4.2.3 Impact of dataset correlation

To investigate the impact of dataset differences on the OD attack, we designed two groups of experiments. The first group is set under the condition where the original dataset and the hijacking dataset are unrelated, using CIFAR-10 as original task and SVHN as the hijacking task. The second group is set under the condition where the original dataset and the hijacking dataset are related, using CIFAR-100 as the original task and CIFAR-10 as the hijacking task. Moreover, both groups is set to IPC $= 50$.

To provide a more intuitive illustration of the differences among the CIFAR-10, CIFAR-100 and SVHN datasets, we visualize the distribution variations across these datasets using t-SNE (as shown in Fig. 7). It is evident that, in the first group of experiments, SVHN and CIFAR-10 exhibit larger distribution differences, whereas in the second group, CIFAR-10 and CIFAR-100 show similar distributions. Furthermore, Fig. 8 presents the results under the two different settings. Fig. 8 (a) corresponds to the case where the datasets are unrelated, while Fig. 8 (b) corresponds to the case where the datasets are related. In both groups the ASR exceeds $97\%$, and the utility is comparable to that of the clean model. Notably, the ASR in Fig. 8 (b) is slightly higher than Fig. 8 (a), which is attributed to the high similarity between the CIFAR-10 and CIFAR-100 datasets. Through our experiments, we demonstrate that the OD attack exhibits strong attack performance regardless of whether the hijacking dataset is related to the original dataset, highlighting the generalization capability of the OD attack.

## 5 Discussion

In realistic scenarios, victims could employ third-party distilled datasets from open-source platforms (e.g., Hugging Face [1], Kaggle [2]) or purchased from external providers to train or fine-tune models. However, these victims are typically unaware that such datasets could contain embedded malicious tasks. This issue is especially pronounced following an OD attack, as the distilled datasets preserve the visual characteristics of the original task, making the presence of hijacking much harder to detect. Consequently, while distilled datasets can lower training costs, they also introduce risks of model hijacking and, more critically, potential legal liabilities.

## 6 Conclusion

In this paper, we introduce OD attack, a novel model hijacking attack method. OD attack integrates model hijacking with dataset distillation, leveraging distilled osmosis samples to significantly reduce the requirement of poisoned samples. We evaluate the OD attack across multiple datasets and model architectures. The experimental results demonstrate that OD attack can successfully execute the hijacking task while minimizing the impact on the performance of the original task. We hope that the OD attack serves as a cautionary example to emphasize the security risks that model hijacking poses to dataset distillation and urge caution in using third-party or unverified synthetic datasets.

---

[1]https://huggingface.co/datasets/devrim/dmd_cifar10_edm_distillation_dataset/tree/main
[2]https://www.kaggle.com/datasets/ericdeuber/nhl-2nd-period-and-final-scores

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

## ETHICS STATEMENT

We hereby declare that our research was conducted in accordance with the ICLR 2026 Code of Ethics. We affirm that human authors are ultimately responsible for validating all claims, results, and contributions presented in this paper. We confirm the originality of our work. This manuscript has not been previously published and is not under consideration elsewhere. We have not inserted any hidden prompts or other attempts to manipulate the peer review process. We have no conflicts of interest to disclose.

## REPRODUCIBILITY STATEMENT

To facilitate the reproducibility of our work, we provide the following resources: The source code for implementing our proposed method and experiments is available at https://anonymous.4open. science/r/OD-7236/. The datasets used in this study are publicly available. Comprehensive details on experimental setups are provided in this paper, including hyperparameters, model architectures, and computational environment specifications.

## A APPENDIX

### A.1 THE USE OF LLMs STATEMENT

We declare that in the preparation of this manuscript we only used LLM to refine grammar and phrasing, as well as to generate suggestions for potential revisions. All content and conclusions presented in this paper were independently determined by the authors and have been verified by all authors. Furthermore, all outputs generated by the LLM were carefully reviewed by the authors. In addition, we did not provide any third-party confidential information or materials intended solely for review purposes to any models.

### A.1.1 IMPACT OF THE TRAJECTORY LOSS.

During the distillation process, we employed a training trajectory matching method to ensure that the distilled osmosis samples retained the features of the hijacking samples. To verify the necessity of incorporating this training trajectory matching approach and its potential to enhance the performance of the OD attack, we conducted ablation experiments. In these experiments, we selected CIFAR-10 as the original dataset and MNIST as the hijacking dataset. The results in Fig. 6 clearly demonstrate that the ASR of the model trained with samples that have the training trajectory information is significantly higher than that of the model trained without it. This indicates that adopting training trajectory matching is crucial for the OD attack. Furthermore, as shown in Fig. 6, incorporating training trajectory matching does not influence the model's utility, suggesting that our attack imposes negligible effects on the original task while demonstrating exceptional stealth capabilities.

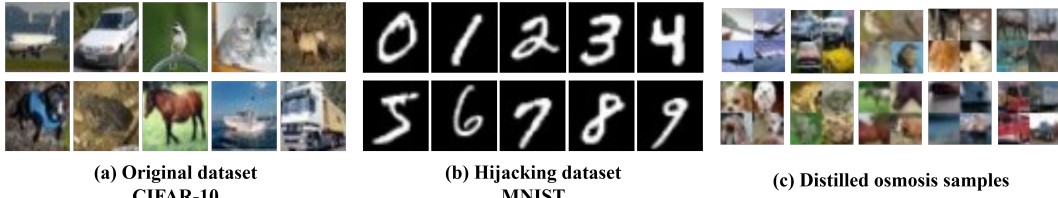

| (a) Original dataset | (b) Hijacking dataset | (c) Distilled osmosis samples |
| CIFAR-10 | MNIST | |

Figure 5: Visualization of the output of OD attack. Figure (a) shows the samples of the Original dataset, figure (b) shows the samples of the hijacking dataset, and figure (c) shows the distilled osmosis samples.

Table 1: Notations

| Symbols | Definitions |
|---|---|
| $x_o$ | Original samples |
| $x_h$ | Hijacking samples |
| $x_c$ | Osmosis samples |
| $x_{c\_syn}$ | Distilled osmosis samples |
| $\mathcal{D}_o$ | Original dataset |
| $\mathcal{D}_h$ | Hijacking dataset |
| $\mathcal{D}_{c\_syn}$ | Distilled osmosis dataset |
| $\mathcal{F}_{(x_c)}$ | Feature extractor for osmosis samples |
| $\mathcal{F}_{(x_h)}$ | Feature extractor for hijacking samples |

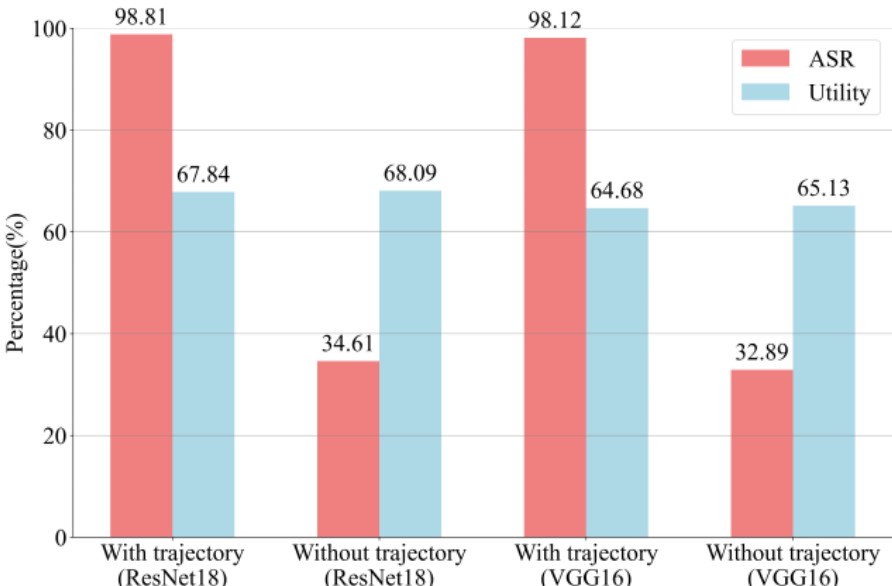

Figure 6: The results of the ablation study on whether training trajectory matching is used during the distillation stage, with CIFAR-10 as the original dataset, MNIST as the hijacking dataset, and IPC = 50.

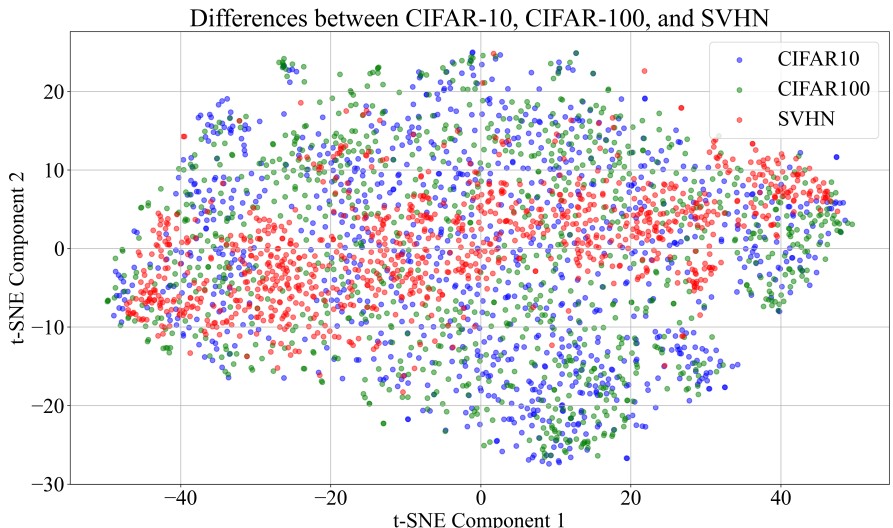

Figure 7: Visualization of comparing different datasets using t-distributed Stochastic Neighbor Embedding (t-SNE)

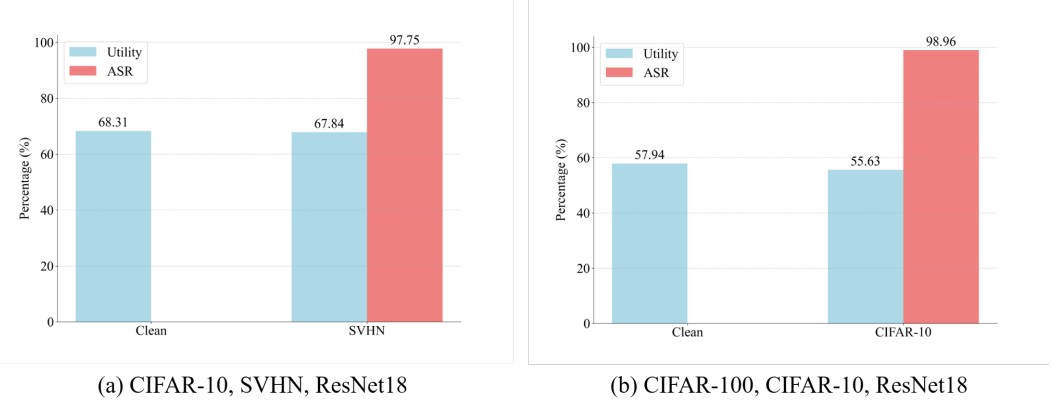

(a) CIFAR-10, SVHN, ResNet18          (b) CIFAR-100, CIFAR-10, ResNet18

Figure 8: Experimental results on dataset correlation

