# OpenReview forum: "Don't Trust any Distilled Dataset! Model Hijacking with the Fewest Samples"
_ICLR.cc/2026/Conference — ICLR 2026 Conference Withdrawn Submission_

### Official Review · Reviewer_2cV8 · 2025-10-15

**Soundness:** 2
**Presentation:** 3
**Contribution:** 3
**Rating:** 2
**Confidence:** 4

**Summary:**

This paper introduces OD (Osmosis Distillation) attack, which embeds an additional “hijacking” task into the victim model while preserving original-task utility. The method relies on visual + semantic losses during generation, patch-based distillation with soft labels, and trajectory-matching to preserve the hijacking signal in a very small distilled set (e.g., IPC ≈ 50).

**Strengths:**

1. This paper is well-organized and easy to read.
2. This paper enhances the stealth of the attack by reducing the amount of data and maintaining high utility on source tasks.
3. The experimental results demonstrate that the attack transferability across different model architectures within the same task.

**Weaknesses:**

1. The manuscript does not clearly bridge the gap between “model contains a hidden classification ability” and “this constitutes a practical, exploitable real-world threat.” As a result, the severity of the threat in practice remains speculative. Specifically:
   * The paper argues stealthiness because the original task utility remains high, but leaves open how an attacker benefits operationally from a model that silently encodes a second task (e.g., remote extraction, covert telemetry, or third-party abuse)
   * Experiments are conducted on common small vision benchmarks (CIFAR/MNIST/SVHN/Tiny-ImageNet). While these validate feasibility, they do not demonstrate the attack at scales or task modalities where legal/ethical harm would actually occur.

2.  The OD formulation and experiments assume that the hijacking task shares the same (or mappable) output space as the source task, raising two concerns:
   * The attack is primarily demonstrated for same-type tasks (classification → classification) and requires an explicit mapping between source and hijack labels. The paper does not experimentally or analytically evaluate the difficulty of cross-task attacks (e.g., classification → object detection, regression tasks), where output dimensionality or semantics differ. The extent to which OD generalizes across heterogeneous task types (and architectures) is therefore unclear.
   *  When the hijacking label space is larger than the source label space, the paper does not provide a principled mapping or capacity analysis, which leaves open whether OD is practical for richer hijacking tasks.

In general, without a clearer exploitation narrative or a demonstration on a more realistic downstream task, I consider this work an interesting proof-of-concept rather than evidence of an immediate, practical security crisis. And the application limitations reduce the generality of the claims. If OD only works for a narrow class of same-task, same-output-space scenarios, that should be stated more clearly.

**Questions:**

None

---

### Official Review · Reviewer_Xm82 · 2025-10-27

**Soundness:** 3
**Presentation:** 2
**Contribution:** 2
**Rating:** 4
**Confidence:** 3

**Summary:**

This paper proposes a novel model hijacking method that leverages dataset distillation to embed malicious hijacking tasks into victim models using very few samples. The method employs a U-Net–based Transporter to generate visually and semantically blended “osmosis samples” between benign and hijacking datasets, followed by a distillation stage that compresses these samples while preserving hijacking effectiveness through key patch selection, label reconstruction, and training trajectory matching. Extensive experiments across CIFAR-10/100, MNIST, SVHN, and Tiny-ImageNet demonstrate high attack success rates (ASR) (>95%) while maintaining high utility (model accuracy on original task). The authors argue that this exposes new security risks when using third-party synthetic datasets in transfer learning.

**Strengths:**

- This work brings attention to a real and under-explored risk in the rapidly growing use of third-party distilled datasets for transfer learning. Highlighting this vulnerability is of clear significance to the community.
- The two-stage Osmosis + Distillation pipeline is conceptually elegant and technically plausible, combining semantic-visual embedding with trajectory matching for stealthy attack transfer.
- Systematic experiments are performed on multiple common datasets (CIFAR-10, CIFAR-100, MNIST, SVHN, Tiny-ImageNet) and two architectures (ResNet18, VGG16), examining not just the attack’s effectiveness, but also its stealth, sample efficiency, and transferability.
- The ablation (see Figure 6) concretely demonstrates the value of their trajectory loss for improving attack potency without degrading benign accuracy.

**Weaknesses:**

1. The paper raises a critical alarm about model hijacking but does very little to discuss possible detection or defense strategies, mitigation mechanisms, or even basic analyses on how one could screen for osmosis/hijacking samples. This significantly lessens its practical impact for practitioners.
2. Only CAMH (He et al., 2025) is compared. Other modern data poisoning, model hijacking, or backdoor-in-distillation baselines (e.g., Liu et al., NDSS 2023) are missing, which weakens claims of superiority.
3. For Eq. 6, $\phi_{h}$ is vaguely described as a “human observer” but this is undefined and impractical; the implementation details of $\phi_h$ and whether it is simulated, learned, or manual are absent.
4. The paper lists relevant prior work (especially Salem et al. NDSS 2022, Si et al. USENIX 2023), but does not cite or discuss two recent, closely related studies:
   - Chung et al. (2024), “Rethinking Backdoor Attacks on Dataset Distillation: A Kernel Method Perspective” — this offers a theoretical basis for understanding dataset distillation vulnerabilities and should be discussed in Section 2 and referenced when motivating the kernel/matching used here.
   - Ge et al. (2024), “Hijacking Attacks against Neural Network by Analyzing Training Data” — this study’s attack methods and analysis of attack mechanisms are relevant benchmarks and should be positioned against OD.

**Questions:**

1. Is there any baseline detection mechanism (e.g., outlier detection, dataset forensics, pattern analysis) that can partially or fully mitigate the OD attack? Have the authors attempted any preliminary analysis (visual/algorithmic) of the osmosis samples’ separability from real data?
2. In Eq. 6, what precisely constitutes $\phi_h$? Is it a perceptual metric, a learned classifier simulating human preference, or a simulated metric? Please provide implementation details.
3. For fine-grained datasets (e.g., CIFAR-100, Tiny-ImageNet), what is the minimum number of per-class hijacking samples required to achieve a given ASR, and how does it scale with class count? Can the authors offer any theoretical or empirical guidance?

---

### Official Review · Reviewer_J4ba · 2025-11-01

**Soundness:** 2
**Presentation:** 2
**Contribution:** 2
**Rating:** 4
**Confidence:** 4

**Summary:**

In this manuscript, the authors introduce Osmosis Distillation (OD) attack, a novel model hijacking attack method. OD attack integrates model hijacking with dataset distillation, leveraging distilled osmosis samples to significantly reduce the requirement of poisoned samples. The authors' experiments demonstrate that OD attack can successfully execute the hijacking task while minimizing the impact on the performance of the original task.

**Strengths:**

- Leveraging dataset distillation enables significant gains in both the efficiency and the stealthiness of the attack.

**Weaknesses:**

- The comparison in Fig. 3 may not be fully suitable, as CAMH is not designed to optimize attack success under extremely low-sample situation. I recommend the authors additionally report CAMH’s best achievable performance. This would help clarify whether the proposed method achieves efficiency at the cost of attack success or model utility.
- All experiments are restricted to ResNet and VGG16. This narrow architectural scope weakens claims about transferability and generality.
- The manuscript does not report the time cost of dataset distillation. While the proposed method substantially reduce the requirement of sample size at the attack stage, it is unclear how much additional cost is required during the training stage.
- It is unclear how the method handles cases where the hijacking dataset and the original dataset have different class numbers, especially for hijack has more classes than the original.
- The proposed method can ensure stealthiness, as distilled osmosis samples exhibit a high degree of visual similarity to original samples. However, beyond human visual inspection, are there potential defense mechanisms for detecting such samples? For example, could OOD detectors, feature-space anomaly detection, or confidence/energy-based scoring be effective in filtering distilled or hijacking samples?
- Would stronger fine-tuning stragies beyond vallina cross-entropy (e.g., alternative loss functions, label smoothing, data augmentation) can reduce attack success rate?

**Questions:**

Please see the weaknesses.

---

### Official Review · Reviewer_MBxZ · 2025-11-01

**Soundness:** 2
**Presentation:** 3
**Contribution:** 2
**Rating:** 4
**Confidence:** 4

**Summary:**

This paper introduces a novel model hijacking attack method called Osmosis Distillation. The proposed approach combines dataset distillation with data poisoning–based model hijacking, enabling the embedding of malicious tasks into benign distilled datasets using only a few attack samples. The OD attack consists of two stages. In the Osmosis stage, a U-Net-based Transporter model fuses the information of original and hijacking samples. By jointly optimizing visual loss and semantic loss, the Transporter generates osmosis samples that are visually similar to the original samples but semantically close to the hijacking samples. In the Distillation stage, key patch selection, soft label reconstruction, and training trajectory matching are applied to the osmosis samples to obtain a compact yet high-fidelity Distilled Osmosis Dataset. When a victim model is fine-tuned on this DOD, it performs normally on the original task but activates the attacker-defined hidden task when encountering specific malicious inputs. Experimental results show that OD achieves over 97% attack success rate on CIFAR-10, CIFAR-100, SVHN, MNIST, and Tiny-ImageNet, while maintaining nearly unchanged performance on the original task, outperforming the current state-of-the-art CAMH attack.

**Strengths:**

- The paper is the first to identify a realistic and novel threat model by combining dataset distillation with model hijacking, exposing previously unexplored vulnerabilities in the distillation pipeline.
- Extensive experiments across multiple datasets and architectures (e.g., ResNet-18, VGG-16) demonstrate that the OD attack achieves very high Attack Success Rates while leaving original-task accuracy virtually unchanged, indicating strong stealthiness.
- Introduces and validates a novel loss term that, for the first time, uses training-trajectory alignment as an explicit loss in the context of dataset distillation with embedded malicious tasks. This causes victim models fine-tuned on the distilled data to follow learning trajectories similar to those produced by the original osmosis samples, thereby concealing the attack.
- The OD attack requires only a few attack samples to succeed, highlighting that the attack remains effective in resource-constrained settings and substantially lowers the cost and barrier for real-world exploitation.

**Weaknesses:**

- Experiments are conducted only on small-scale vision datasets; the paper lacks evaluation on larger-scale datasets (e.g., ImageNet), leaving questions about scalability and real-world applicability.
- The attack model assumes the victim will use a distilled dataset produced or published by the attacker, which may be an optimistic assumption in some practical deployments.
- Although the paper exposes a real security threat, it does not provide a thorough investigation of detection or mitigation strategies for this class of attacks.
-The paper does not experimentally demonstrate whether OD can evade or be detected by common poisoning detectors, leaving open how resilient the attack is to standard defenses.
- The paper does not analyze what specific semantic information is embedded in the osmosis samples or how that information propagates through the network during fine-tuning; it also lacks feature-level visualizations that could explain the underlying mechanism.

**Questions:**

- Could the authors evaluate OD attack using established backdoor detection techniques to assess whether it can be detected by existing defenses?
- Can OD attack be extended to larger-scale or non-vision domains such as language or diffusion models? If so, how does its performance and stealthiness vary across modalities?
- If the Distilled Osmosis Dataset is combined with a fraction of genuine training data (e.g., 10 % real + 90 % DOD), does the Attack Success Rate drop significantly?
- What are the ablation results for patch size, N (number of key patches per class), and the image-concatenation strategy? How do these hyperparameters influence the DOD size and the overall attack effectiveness?

---

### Note · Authors · 2025-11-12

I have read and agree with the venue's withdrawal policy on behalf of myself and my co-authors.